# The Co-Design/Co-Development and Evaluation of an Online Frailty Check Application for Older Adults: Participatory Action Research with Older Adults

**DOI:** 10.3390/ijerph20126101

**Published:** 2023-06-10

**Authors:** Bo-Kyung Son, Takahiro Miura, Ken-ichiro Yabu, Yuka Sumikawa, Dongyool Kim, Weida Lyu, Yingxue Yang, Moeko Tanaka, Tomoki Tanaka, Yasuyo Yoshizawa, Katsuya Iijima

**Affiliations:** 1Institute of Gerontology, The University of Tokyo, Tokyo 113-8656, Japan; miu@iog.u-tokyo.ac.jp (T.M.); yabu@human.iog.u-tokyo.ac.jp (K.-i.Y.); weidalvleo@gmail.com (W.L.); tmk-tanaka@iog.u-tokyo.ac.jp (T.T.); yoshizawa@iog.u-tokyo.ac.jp (Y.Y.); iijima@iog.u-tokyo.ac.jp (K.I.); 2Institute for Future Initiatives, The University of Tokyo, Tokyo 113-0033, Japan; 3Department of Geriatric Medicine, Graduate School of Medicine, The University of Tokyo, Tokyo 113-8655, Japan; 4Human Augmentation Research Center (HARC), National Institute of Advanced Industrial Science and Technology (AIST), Kashiwa 277-0882, Japan; 5Research Center for Advanced Science and Technology (RCAST), The University of Tokyo, Tokyo 153-8904, Japan; 6Division of Health Sciences and Nursing, Graduate School of Medicine, The University of Tokyo, Tokyo 113-0033, Japan; sumi-tky@g.ecc.u-tokyo.ac.jp (Y.S.); sagefemme.dansante.m@gmail.com (M.T.); 7Department of Agribusiness Management, Faculty of International Agriculture and Food Studies, Tokyo University of Agriculture, Tokyo 156-8502, Japan; dongyool.kim30@gmail.com; 8Graduate School of Education, The University of Tokyo, Tokyo 113-0033, Japan; yuki123yang@hotmail.com

**Keywords:** online frailty check application, older adults, co-design, co-development, reliability, participatory action research

## Abstract

Frailty is an age-related condition characterized by a decline in physical capacity with an increased vulnerability to stressors. During the COVID-19 pandemic, there was considerable progression in frailty in older adults. Therefore, an online frailty check (FC) is required for continuous screening, especially acceptable to older adults. We aimed to co-design/co-develop an online FC application with FC supporters who were facilitators in a pre-existing onsite FC program in the community. It consisted of a self-assessment of sarcopenia and an 11-item questionnaire assessing dietary, physical, and social behaviors. Opinions obtained from FC supporters (median 74.0 years) were categorized and implemented. The usability was assessed using the system usability scale (SUS). For both FC supporters and participants (n = 43), the mean score was 70.2 ± 10.3 points, which implied a “marginally high” acceptability and a “good” adjective range. Multiple regression analysis showed that the SUS score was significantly correlated with onsite–online reliability, even after adjusting for age, sex, education level, and ICT proficiency (b = 0.400, 95% CI: 0.243–1.951, *p* = 0.013). We also validated the online FC score, which showed a significant association between onsite and online FC scores (R = 0.670, *p* = 0.001). In conclusion, the online FC application is an acceptable and reliable tool to check frailty for community-dwelling older adults.

## 1. Introduction

Frailty is a complex age-related clinical condition characterized by a decline in physiological capacity across several organ systems [1,2]. Frailty exposes the individual to a greater risk of multiple adverse health outcomes such as loss of mobility, falls/fractures, hospitalization, and early mortality [3,4,5,6,7]. Three important factors have been suggested in the concept of frailty. First, frailty is multidimensional with physical, cognitive, and social factors. Second, although its prevalence does increase with age, frailty is an extreme consequence of the normal aging process. Third, it is reversible, which means that an individual can prevent or slow its progression [8]. Therefore, effective strategies that target the prevention and management of frailty in an aging population will probably reduce the condition’s burden at both the individual and the health system levels.

In Japan, to screen frailty in community-dwelling older adults, we developed a frailty check (FC) program. This program is based on evidence from a large-scale longitudinal cohort study of older adults in Kashiwa City who did not require long-term care needs [9]. The FC program consists of a self-assessment for sarcopenia (Yubi-wakka test) [10] and an 11-item self-reporting questionnaire. Particularly, this simple check version is designed for self-awareness to motivate people to change their lifestyle to incorporate good nutrition [11], physical activity [12,13], and social participation [14], which are the three pillars from the perspective of preventing frailty. Notably, the FC program is characterized by a citizen-centered action handled by FC supporters who are community-dwelling older volunteers. The FC program was conducted in groups at public halls and community centers easily accessed by older adults. FC supporters encourage participants to improve their lifestyles and practice frailty prevention [15]. From April 2015 to February 2020, a total of 8855 community-dwelling older adults participated in the FC program in 47 local government prefectures. Thus, FC supporters are important stakeholders in frailty prevention in the community.

Recently, the coronavirus disease 2019 (COVID-19) pandemic caused by severe acute respiratory syndrome coronavirus 2 (SARS-CoV-2) resulted in restrictions on going outdoors, which led to a decrease in physical activity among older adults, which may have led to sarcopenia and frailty [16,17,18,19,20]. Our recent study also found a significant decrease in the trunk muscle mass in older adults (one of the risk factors for postural instability and falls) immediately after the pandemic’s first wave (April–May 2020) [20]. Furthermore, we found that appendicular muscle mass and grip strength continued to decrease for 1 year during the COVID-19 pandemic [21]. The frailty caused by the COVID-19 pandemic became known as corona-frailty. Thus, to monitor frailty continuously during the COVID-19 pandemic, an online FC application was needed.

Recently, studies have reported online frailty assessments for older adults [22,23] or health professionals [24]. For example, in community settings, using an online version of the FRAIL scale (a simple questionnaire including five items: fatigue, resistance, ambulation, illness, and weight loss), participants aged 60 years or older were able to understand the questions and answer using tablets [22]. A significant correlation with SARC-F was also observed. However, another study showed that only 35% of older adults accepted a digital approach for conducting health assessments or accessing assessment results [22].

To be acceptable for older adults based on the established onsite FC program, we co-designed/co-developed an online FC application with FC supporters through participatory action research (PAR). In this study, we aimed to examine the usability of online FC focusing on reliability, defined as the consistency of pre-existing onsite FC, and online communication. We also validated the online FC scores with those of onsite FC.

## 2. Materials and Methods

### 2.1. Older Adults: Frailty Check Supporters and Participants

In this study, 32 FC supporters were involved in the development and implementation of the online FC application. The FC supporters were community-dwelling older adults who had facilitated the onsite FC program in Bunkyo-ku and Nishitokyo-shi in Tokyo, Japan. Additionally, we recruited 20 community-dwelling older adult participants who had prior experience with the onsite FC program in Nishitokyo-shi, Tokyo, Japan. This study protocol was reviewed and approved by the University of Tokyo institutional review committee (approval number: 21-190). Written informed consent was obtained from all FC supporters and participants by researchers before patient interview or recruitment.

### 2.2. Study Design: Participatory Action Research

A flow diagram of the development and implementation process is shown in Figure 1. To develop and implement the online FC application, we considered PAR to be the most appropriate method for several reasons: (1) frailty screening and prevention are critical issues in the community, (2) its participatory nature involves FC supporters in the research from the beginning to the end, and (3) it involves research in action. However, there is limited research using PAR as a technological tool for frailty screening, especially in older adults [25,26].

To co-design the application, referring to recent studies [27,28,29], focus group interviews and mock tests were conducted with FC supporters, and self-efficacy questionnaires were assessed before and after the mock test. After online FC implementation, we conducted surveys on the onsite–online reliability, online communication, and usability of the user interface (Figure 1a).

#### 2.2.1. Online Frailty Check Application: Device and Contents

The online FC was conducted via the FC application using a tablet device by joining a video conference room with six seats (FC supporter: participants = 1–2:1–5). This ratio was intended to enable FC supporters to adequately attend to the participants (Figure 1b). In this video conference room, an FC supporter could share the questionnaire with the participants on screen and grasp the state of participant responses. Following the FC supporters’ instructions, participants would input data or answer questions using an electronic pencil.

The online FC program consisted of three sections: basic information (age, sex, body weight, and height); the Yubi-wakka test (a sarcopenia test with video instruction [10]) and an 11-item FC questionnaire (yes/no, validated questionnaires including questions on nutrition, oral and physical function, and social activity). Inputted data were summarized at the end, which the participants could confirm prior to submission. This design was also used in the onsite FC program. Each section of the 11-item questionnaire is addressed in Appendix A, and a movie of the online FC implementation is also provided as Appendix A.

#### 2.2.2. Development and Refinement of Application

To facilitate the smooth execution of the experiment and the collection of authentic and reliable data, the process was divided into two main parts. To collect primary data, we began by conducting online FC briefing sessions and conducted several meetings for user experience sharing. The primary data were extracted from focus group interviews, discussions, and observations, as outlined in Figure 1. Therefore, we tried to ensure the validity and reliability of our research by triangulating multiple data sources [30]. User experience meetings for the prototype application were held over 2 days, and a total of 32 FC supporters participated. Subsequently, we improved the application based on the opinions and feedback received.

We explored the challenges that participants encountered in using the application. Therefore, we followed previous studies on application development and action research based on grounded theory [31,32,33].

Data analysis was conducted using the following steps: First, we coded the interview data using first-order codes: (1) reliability, (2) social interaction, and (3) user-friendliness. Second, we added the frequency to indicate the importance of each code. After coding and categorization, we reviewed and refined each opinion via discussion.

### 2.3. Questionnaires

To examine whether FC supporters were empowered by PAR, we assessed self-efficacy using a validated 16-item questionnaire on the general self-efficacy scale (GSES) consisting of three categories: positive behavior, non-anxiety about failure, and social positioning of ability [34]. The total possible score on the GSES is 16 points. Based on the results of a previous study in the Japanese population (n = 278), it was suggested that 10–16 points indicate higher self-efficacy, 4–9 points indicate standard self-efficacy, and 0–3 points indicate lower self-efficacy [34].

For both the FC supporters and participants, we evaluated the user interface of the online FC application using relevant questionnaires. To assess the onsite–online reliability (3 items) and interaction quality (4 items), we utilized modified questionnaires on telehealth usability (Appendix A) [35]. For quantitative data, we scored the system usability scale (SUS). The SUS has been extensively used in previous user research studies and demonstrated good psychometric properties [36]. Better usability was indicated by higher SUS scores, ranging from 0 to 100 [37]. A SUS score of 68 is the center of the Sauro–Lewis curved grading scale, which is one of analyzing points of usability [37,38]. The SUS consists of 10 items, each with five steps anchored with “strongly disagree” and “strongly agree”. It is a mixed-tone questionnaire in which the odd-numbered items have a positive tone, and the even-numbered items have a negative tone.

### 2.4. Analysis

The differences between FC supporters and participants were examined using Student’s *t*-test or the Mann−Whitney U-test after confirming normal distribution using the Shapiro–Wilk test. We examined the effect on self-efficacy before and after the mock test using a Wilcoxon rank-sum test, after confirmation of normal distribution using the Shapiro–Wilk test. Cronbach’s alpha was calculated to appraise the scale’s internal consistency. To analyze the relationship between the SUS and the related independent variables, as well as between online and onsite FC results, we applied Pearson’s or Spearman’s correlation analysis. Multiple regression analysis was performed using SUS as the dependent variable and age, sex, education, ICT proficiency, reliability, and interaction quality as independent variables. Data were analyzed using IBM SPSS Statistics version 26 (IBM Japan, Tokyo, Japan). Statistical significance was set at *p* < 0.05.

## 3. Results

### 3.1. Frailty Check Supporters and Participants

The basic characteristics of the FC supporters and participants are presented in Table 1. Of the 12 FC supporters, 7 were men. Compared with FC supporters, the participants were significantly older and had lower education levels, while subjective health and well-being did not differ between the two groups. The subjective proficiency of ICT tended to be low for the participants (*p* = 0.082). Among participants, two major reasons for participation in the online FC were identified: (1) interest in the online FC program (40%) and (2) the need for continuous frailty checks (40%) owing to the cessation of the onsite FC program during the COVID-19 pandemic (data not shown).

### 3.2. Development and Refinement of the Application

The opinions collected from the focus group interviews and the refinement process are presented in Table 2. Fifty-five detailed opinions were obtained and classified into five categories based on keywords such as reliability, social interaction, and user-friendliness. The largest number of opinions (n = 18) were related to reliability such as blue/red stickers. For example, to improve familiarity with the online questionnaire format, the onsite format of the 11-item self-reported questionnaire was adapted and modified to display one question at a time on a single page. In addition, a confirmation page was added so each participant could reconfirm all the responses. Furthermore, FC supporters expressed difficulties in online communication, such as identifying the speaker and capturing their reaction. To enhance and encourage communication, we implemented several measures, including spotlighting the person speaking, providing an explanation of the system, and removing the daily topic section before the check.

FC supporters expressed difficulty in conducting the Yubi-wakka test using the screen. To address this challenge, we created a video explaining the test together with the supporters and embedded it into the application to provide an accurate and consistent explanation similar to the onsite FC.

Furthermore, several opinions on user-friendliness were collected, and improvements based on this feedback were implemented: (1) using a stylus, (2) enlargement of font size, and (3) introducing a button-pressing simulator. In particular, we found that including a stylus significantly improved usability by solving dry fingers and touch sensitivity issues, especially for older adults. We also received suggestions on font size enlargement and improving readability. A button-pressing simulator was also included for participants new to tablets, bridging the gap between physical interfaces and touch-based interaction for a smoother transition. Our study involved 52 participants, ensuring a diverse sample size to capture various perspectives and usability issues

### 3.3. Self-efficacy of Frailty Check Supporters and Usability Evaluation

In the co-design and co-development of the online FC application, we investigated whether the engagement of FC supporters in PAR was empowered, using a validated self-efficacy assessment. As shown in Table 3, a slight but not significant increase in the total GSES score was observed after the mock test (before: 12.0 vs. after: 13.0). However, among the three GSES categories, we noticed a trend of improvement in the non-anxiety of failure category, as the proportion who answered “no” regarding (Q5) “I am more concerned about small failures than others all the time”, tended to increase after the mock test (before: 63.0% vs. after: 77.8%, *p* = 0.094). Additionally, in the category of the social positioning of ability, the proportion of “yes” responses to (Q3) “there are areas where I have better knowledge than friends,” significantly increased (*p* = 0.031) in comparison to results from before the mock test. Similarly, the proportion of “yes” responses to (Q1) “I have better ability than friends,” also increased (*p* = 0.063).

The overall system usability of the online FC application was evaluated using the SUS, a widely used questionnaire with an acceptability and adjective range scale (Figure 2). We found that the average SUS scores of both FC supporters and participants were similar and in the “marginally high” section of the acceptability range. Additionally, the scores were above the average (68 points [38,39]) of the adjective ranges, which implies that the developed FC application is suitable for older adults.

Next, we examined the overall SUS score and factors that affect the user interface. As shown in Table 4, we found that age, education level, subjective ICT proficiency, reliability, and interaction quality were significantly associated with the SUS score (*p* < 0.05). Furthermore, when we assessed the multiple (linear) regression analyses, we observed a significant correlation between the SUS score and the reliability of the online FC application after adjusting for covariates, including age, sex, education duration, and ICT proficiency (Table 5). On the other hand, the interaction quality did not display a significant association with the SUS score.

We further validated the results of the online FC program by comparing them against those of the onsite FC, which were conducted within 4 months of each other. Pearson’s correlation coefficient revealed a significant association between the onsite and online results (*p* = 0.001) (Figure 3).

## 4. Discussion

In this study, we developed an online FC application for community-dwelling older adults delivered via video conferencing, during the COVID-19 pandemic. To enhance its acceptability in older adults through PAR, we co-designed/co-developed it with stakeholders named FC supporters who are older community-dwelling volunteers facilitating the pre-existing onsite FC. This study aimed to examine the usability and validity of the online FC application.

To the best of our knowledge, this is the first online application for frailty checks designed by older adults, for older adults. To achieve our purpose, we utilized a PAR approach [33,40,41], consisting of focus group interviews and a mock test. Given that the FC program is an independent civic activity, and FC supporters were not simply users but also operators who should facilitate both onsite and online FC, they were involved in all steps of this study, from the application design, e.g., what type of technology is acceptable for older adults and how should it be delivered to participants, to the step of implementation of online FC with participants, which are finally attributed to their empowerments. Based on this observation, it is conceivable that a participatory design might be a useful methodology when developing digital healthcare interventions for older adults.

Previous studies for online FC assessment demonstrated that only one-third of participants adopted a digital approach including computers, smartphones, and tablets to self-assess frailty status, although most participants agreed on the usefulness of online assessment to understand their health status. In this study, to dissolve this concern, we explored the use of video conferencing through tablets and found it to be a potentially useful online tool for older adults, as they can communicate and answer the questionnaire together during the frailty check. Consistent with our assumption, video conferencing has been proposed as a medium for digital health intervention that can be used for communication, providing care and support, and enhancing health status, such as home-based tele-exercise for patients with chronic diseases [42,43]. However, the efficiency and cost-effectiveness of video-conferencing technology should be examined in future studies.

In this study, we suggested the use of continuous and effective FC in the community using a hybrid system linking onsite (every 6 months) and online FC (e.g., once or twice between 6 months) by understanding benefits (e.g., no risks of falling going to the appointment) and disadvantages (e.g., not going to meet people in person, appointments give some people a reason to leave the house) of online FC.

Intriguingly, regarding the self-efficacy of FC supporters, we found that an item for the social positioning of ability significantly changed after the mock test, and a trend of improvement in the non-anxiety of failure was observed. These results suggested that the FC supporters were able to strengthen their own social positioning of ability and relieve anxiety about failure through PAR in the design/development of the application. This highlights the promotive effects on self-efficacy for FC supporters, although the average total score did not significantly change because the total score before the time point was high enough. Furthermore, consistent with a recent study on PAR [44], the empowerment of FC supporters equips them with the confidence to operate the online FC, which subsequently could enhance the usability for all participants.

In this study, we found that the average scores of SUS for the online FC application are 70.2 ± 10.3, which indicated a marginally high rate of acceptability. Consistently, it was demonstrated that SUS scores of community-dwelling older adults (70.9 ± 5.6 years) were 68 in a recent pilot study for the system for assessment and intervention of frailty [45]. Although the SUS is a standardized questionnaire designed to assess perceived usability [36], there is a lack of studies exploring the usability of technology, especially for older adults. Further investigations involving large-scale randomized controlled trials are needed.

The results of this study showed that the onsite–online reliability was significantly associated with SUS scores, even after adjustment for several variables, including sex, age, educational level, and subjective ICT proficiency. This finding suggests that the similarity between the contents and operation of the online FC program and the pre-existing onsite FC program contributes to the safety, satisfaction, and usability of older participants. However, interaction quality was not significantly associated with SUS scores, indicating that several issues need to be addressed in online communication among older adults. For example, audio–video quality (e.g., dim lighting, small voice) should be improved to enhance communication, particularly in terms of social isolation and loneliness, considered the first step of frailty [46].

Finally, we validated the results of online FC assessments by comparing them with those of the onsite FC. The significant similar scores observed between the two could be attributed to the reliability of the application (11 yes/no questions). With this observation, it is suggested that the online FC application could be a reliable tool of onsite FC among older participants, for sustained monitoring of frailty status with a hybrid system and further use for emergency situations such as the COVID-19 pandemic.

### Limitations

Our study had several limitations. The primary focus was on developing the application and conducting a preliminary evaluation of its acceptability, reliability, and usability. This study was not designed to rigorously test the efficacy of the online FC assessment in improving frailty status. Additionally, the use of non-systematic recruitment methods may have introduced a bias toward participants who were more open to this type of intervention. Our design/development was limited to Japanese older adults. To build a generalizable system, an established protocol of PAR involving community stakeholders is needed.

Despite these limitations, our study has several important implications: (1) individual FC supporters and participants may benefit from using the online FC application; and (2) a participatory design approach is a useful methodology for developing a relevant, useful, and acceptable tool for older adult users.

## 5. Conclusions

We co-designed/co-developed an online FC application using a participatory action research approach to create an acceptable tool for older adults to assess their frailty. Our study revealed that the reliability of onsite FC is an important determinant affecting the usability and satisfaction of the online FC application. Furthermore, it is conceivable that the enhanced empowerment of FS supporters may lead to high usability scores in participants, a win–win situation. Further investigations are needed to rigorously test the efficacy of the developed online FC application with larger sample sizes.

## Figures and Tables

**Figure 1 ijerph-20-06101-f001:**
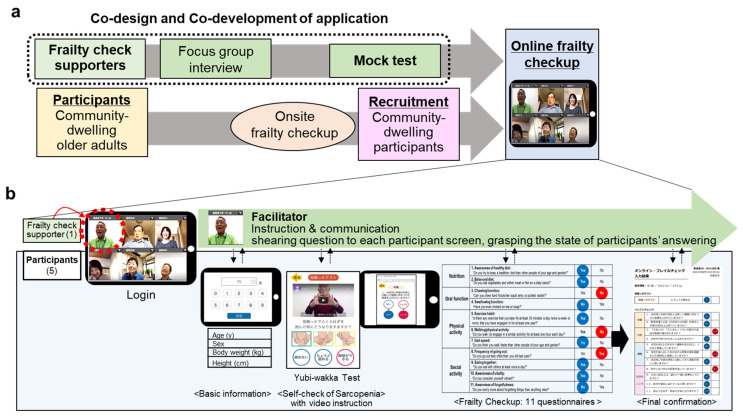
Development and implementation of an online frailty check application. Co-design and co-development of the application were performed with frailty check (FC) supporters, followed by online FC implementation and surveys (**a**). The contents of the online FC and the role of the FC supporters as facilitators are shown (**b**).

**Figure 2 ijerph-20-06101-f002:**
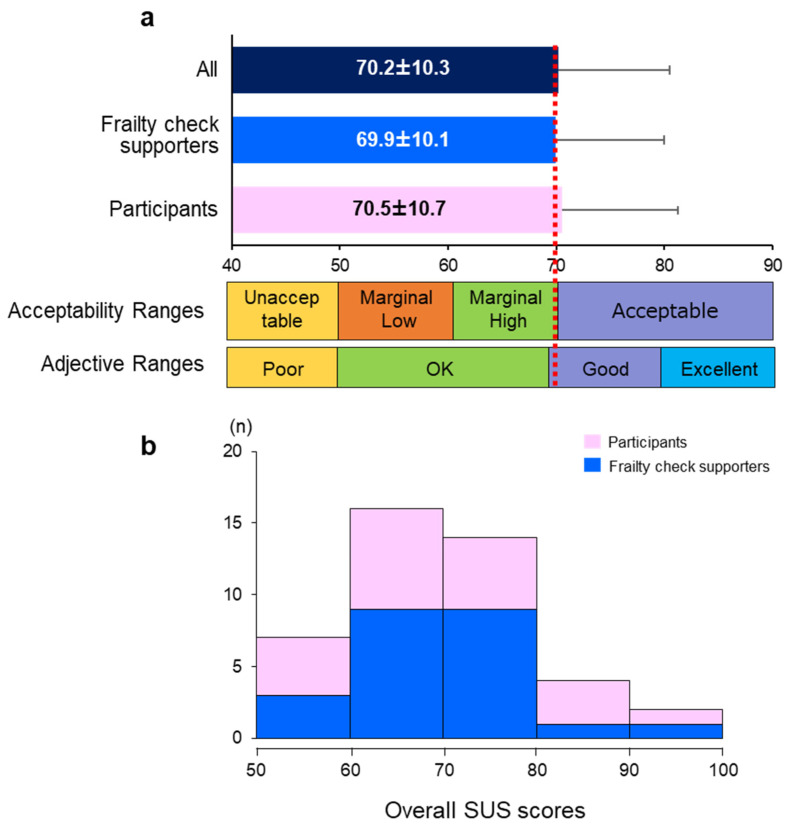
System usability scale (SUS) scores of the online frailty check application (**a**,**b**). Error bars represent the standard errors of each SUS score. The red dotted line represents a rough standard of the adjective ratings (68 points); n = 43 (FC supporter n = 23, participants n = 20).

**Figure 3 ijerph-20-06101-f003:**
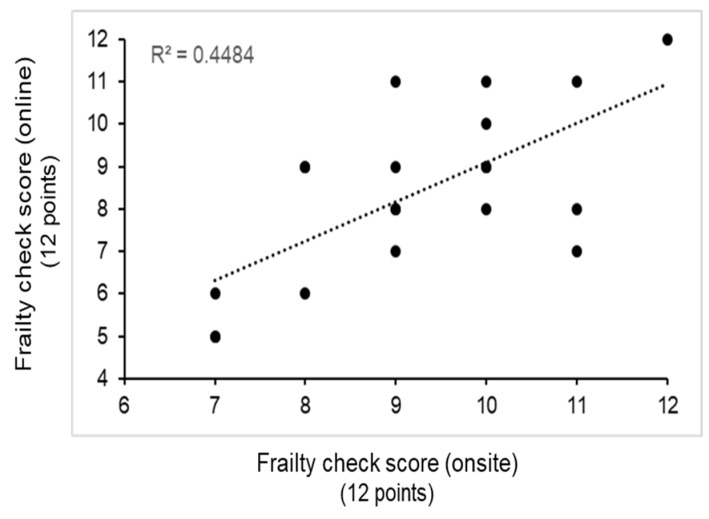
Association between the onsite and online frailty check scores of the participants (n = 20). Pearson’s correlation analysis (0.670, *p* = 0.001).

**Table 1 ijerph-20-06101-t001:** Basic characteristics of the frailty check supporters and participants.

	Frailty Check Supporters	Participants	*p ^#^*
n	32	20	
Men/Women	20/12	13/7	0.855
Age (y)	74.0 (67–86) *	82.0 (70–91)	0.001
Education level (longer than 14 years, %)	23 (71.9%)	7 (35.0%)	0.009
Resident period (years)	40.5 ± 16.0 **	35.3 ± 17.2	0.277
Living alone (%)	7 (21.9%)	6 (30.0%)	0.752
Subjective health (healthy status %)	32 (100.0%)	19 (95.0%)	0.202
Subjective well-being (10 points)	8 (3.0–10.0)	8 (5.0–10.0)	0.854
Subjective proficiency in ICT	5.5	5	0.082
(10 points)	(1.0–9.0)	(0.0–10.0)	

* Values are presented as median (min–max), non-normal distribution. ** Values are presented as mean ± standard deviation, normal distribution; # *t*-test or Mann–Whitney test for continuous value with normal distribution or non-normal distribution.

**Table 2 ijerph-20-06101-t002:** Opinions and refinements for the online frailty check application expressed by frailty check supporters.

Category	Opinions (No. of Opinions)	Example for Opinions	Refinement
Reliability	Item entry (1)	“It is difficult for participants to enter the numbers unless FC supporters ask them verbally.”	Preparation of information to be entered before the test
	Progress status (3)	“It would be better to have a display showing that other participants are still answering.”	Addition of a page to check the progress
	Final confirmation page (3)	“It would be better to be able to recognize which questions have not been answered at the end.”	Addition of a confirmation page
	Homogeneity with onsite frailty check (3)	“I want the answer page to be displayed similarly the onsite 11 self-reporting FC questionnaires page.”	Change to a similar design as the onsite questionnaire
	Change of display name (1)	“It would be better to be able to change the display name on the app.”	Indication of the participants’ affiliation in their native language for easy understanding
	Need for training and practice (4)	“I think it would be preferable for supporters to work as a pair so that they could help each other.”	Conduction of self-directed learning activities by the supporters
	Encouragement (1)	“I thought I should support everyone by saying, ‘I couldn’t do it either, but now I can.’”	Addition to the cheering sound effect function
	Provision of frailty check results (2)	“In the onsite frailty check, participants can take home a paper of the results, but what about that during online frailty check?” “Can each participant receive their own data?”	Addition of a printing function and distribution of printed results to users
Social interaction	Identification of speaker (3)	“I want a function/signal designed that allows me to recognize who the speaker is.”	Manualization of communication function
	Tablet camera setting (2)	“My finger hits the camera when I hold the tablet.”	Manualization of settings and facilitation by the supporter
	Online communication (3)	“When we’re online, we can’t properly capture the other side’s reaction, so we talk less often.”	Building intimacy through daily conversation before the measurement
User-friendliness	Button (4)	“Buttons are small and difficult to touch with fingertips. I think it would be easier to do with a stylus.”	Use of a stylus
	Touch screen sensitivity (2)	“Buttons do not respond when my fingers are dry.”	Use of a stylus
	Touch screen skills (4)	“I don’t know how to press the button.”	Instruction provision, Addition of button pressing simulator
	Font (3)	“A larger font is better.”	Use of a larger font size
	Initialization function (1)	“It is better to design the question form so that the users can go back to the previous question.”	Manualization of operation method and clarification of troubleshooting
	Tutorial function (1)	“It would be good to have a button-pressing practice function. Elderly people can’t learn it in one session.”	Addition of button pressing simulator
	Visual design (2)	“I want the design to be relevant to frailty check and the elderly.”	Improvement of layout
Social interaction and Reliability	Timing to watch the Yubi-wakka tutorial video (2)	“Some participants think the video starts automatically, so they need to be verbally informed.”	Addition of a step-by-step tutorial video
	Calling attention (1)	“To prevent accidents, appropriate explanations such as ‘Please sit in a chair during the examination’ are needed.”	Emphasizing in the step-by-step tutorial video, with verbal reminders by the supporter
	Frailty check procedures (1)	“I think it would be easier for participants to 11 self-reporting FC questionnaires, if the supporters read the questions one by one.”	Checking the progress by calling the participant’s name
	Announcement for ending (1)	“It would be better to all say goodbye and then guide participants to press the exit button.”	Instruction manual and verbal reminder
Reliability and User-friendly	Answer format (1)	“It would be better for all participants to answer each question simultaneously.”	Converting one-page forms to one question-per-page form
	Backward function (1)	“On all pages, I want a button to go back to the previous page.”	Addition of backward function
	Display of the test tutorial video (5)	“Please consider making a Yubi-wakka test explanation video in advance and showing it during the check.”	Verbal guidance on how to play the tutorial video

**Table 3 ijerph-20-06101-t003:** Self-efficacy of the frailty check supporters before and after the mock test.

General Self-Efficacy Scale		Before	After	*p* ^‡^
**Total score: Self-efficacy (points/16 points)**		12.0 (5.0–16.0) ^+^	13.0 (6.0–16.0)	0.498
**Category 1: Positive behavior (points/7 points)**		6.0 (1.0–7.0)	6.0 (0.0–7.0)	0.952
Q1	I am confident when I do something. (yes, %)	77.8	77.8	0.375
Q2	I am worried compared to people. (no, %)	55.6	51.9	0.312
Q3	I decide without hesitation when I decide something. (yes, %)	63.0	55.6	0.250
Q4	I think I am a shy person. (no, %)	70.4	70.4	0.375
Q5	I think it’s better to work proactively even in jobs where the results are uncertain. (yes, %)	85.2	88.9	0.500
Q6	I am a person who is willing to do anything. (yes, %)	66.7	77.8	0.125
Q7	I am a person who are not good at actively working. (no, %)	85.2	85.2	0.500
**Category 2: Non-anxiety about failure (points/5 points)**		4.0 (2.0–5.0)	5.0 (1.0–5.0)	0.582
Q1	I often feel dark remembering the mistakes and unpleasant experiences I made in the past. (no, %)	81.5	85.2	0.375
Q2	I often feel that I have failed, after finishing work. (no, %)	92.6	92.6	0.500
Q3	I’m often worried that it won’t work when to do something. (no, %)	77.8	74.1	0.375
Q4	I often can’t get to work since I couldn’t decide what to do. (no, %)	92.6	88.9	0.312
Q5	I am concerned all the time for small failure than others. (no, %)	63.0	77.8	0.094
**Category 3: Social positioning of ability (points/4 points)**		2.0 (1.0–4.0)	3.0 (0.0–4.0)	0.593
Q1	I have better ability than friends. (yes, %)	40.7	55.6	0.063
Q2	I have better memory than humans. (yes, %)	44.4	51.9	0.234
Q3	There are areas where I have a particularly good knowledge than friends. (yes, %)	55.6	74.1	0.031 *
Q4	I think I have the power to contribute to the world. (yes, %)	85.2	88.9	0.375

N = 27. * *p* < 0.05. ^+^ Values are presented as median (min–max). ^‡^ Comparison of values before and after the mock test using the Wilcoxon rank-sum test.

**Table 4 ijerph-20-06101-t004:** Association between the system usability scale (SUS) score and factors affecting the user interface.

	SUS Score	*p*
Age	−0.319 *	0.037
Sex	−0.153	0.328
Education	0.376 *	0.013
ICT proficiency	0.376 *	0.013
Reliability	0.312 *	0.042
Interaction	0.309 *	0.044

N = 43, * *p* < 0.05, the Spearman correlation analysis.

**Table 5 ijerph-20-06101-t005:** Multiple regression analysis of the system usability scale.

	Model 1			Model 2			Model 3		
	β	95% CI	*p*	β	95% CI	*p*	β	95% CI	*p*
Reliability	0.228	−0.343–1.591	0.200	0.326 *	0.004–1.782	0.049	0.400 *	0.243–1.951	0.013
Interaction	0.178	−0.408–1.244	0.313	0.196	−0.289–1.206	0.222	0.139	−0.389–1.038	0.362

Model 1: unadjusted, Model 2: adjusted for age, sex, Model 3: adjusted for age, sex, education level, and ICT proficiency, R = 0.658, R^2^ = 0.433, Durbin–Watson = 2.076. * *p* < 0.05.

## Data Availability

The data in this study are restricted because ethical approval was not sought for public data sharing from the Ethics Committee at the University of Tokyo, and the participants were not informed of possible public data sharing when they provided informed consent. However, data can be made available from a non-author of contact at the Institute of Gerontology, the University of Tokyo (contact via info.frail@iog.u-tokyo.ac.jp) for researchers who meet the criteria for access to confidential data.

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
