# Peer review of "The Co-Design/Co-Development and Evaluation of an Online Frailty Check Application for Older Adults: Participatory Action Research with Older Adults"

_ijerph, 2023, doi:10.3390/ijerph20126101_

Round 1

Reviewer 1 Report

For the abstract, is it possible to provide a more user-friendly definition of Frailty? The journal is “IJERPH” so the readers may be clinicians such as nurses or therapists. The current definition in the abstract is very bench research-sounding and not translational.

The introduction: There is a lack of why this device was created and what issues/limiting factors this device is intending to address. Please add to, and rephrase, frailty to describe a more clinician-friendly definition and take out the reference from 2001. The only limiting factor of frailty that I’m understanding is “physiological capacity due to organ systems.”   The literature review is currently not sufficient for a manuscript to be published, it is lacking critical support to design this device based on a review of technology that is already on the market to address the issue. I’m not understanding what is current, or lacking on the market, to address physiological capacity.  What exactly is this: balance, heart rate, blood pressure, loss of muscle mass, falls?  What is the support in the literature of the development of devices that are technologically sound, feasible, and have preliminary efficacy to address this (outside of Japan), and other issues, of the older adult?  There should also be in the intro any background literature on older adult reports of their acceptance of telehealth/online health sessions, and provide the background work of others to support this device. This should guide then your research question, which the article is also lacking: Please provide a clear research question.

Please also add if this was approved by an ethics board and when subjects consented to their participation.

Line 109: What were the methods that guided your qualitative portion of this? i.e. content analysis, thematic analysis, phenomenology, grounded theory?

There is no description of the General Self-Efficacy Scale questionnaire. How are the scores compiled, what are the ranges of scores, and what are the cut-off scores for interpretation? Is this published?  What are the psychometric properties? Reliability, validity, MCID? If not and it was created, did it go through a pilot phase?  The same comments above apply to the SUS which requires sufficient descriptions. It appears in your graph that 70 is a cut-off of SUS acceptability, but this needs to be stated in the text if there is literature to support this. This usually is listed as the “Measurement” section of a Methods section.

Discussion: There is also not a sufficient discussion of your interpretations of the GSES and the SUS scores if the cut-offs were met for an older adult.

In the discussion, this is the first mention that this is a feasibility study, which cannot be introduced in a discussion.  If this is an aim, add it to the research question and how feasibility was planned to be measured/assessed. If not, please take out the phrasings of feasibility.

Some language is too technical for the clinician readership and lacks translational application.  I asked the authors to address this.

Author Response

Co-Design/Co-Development and Evaluation of an Online Frailty Check Application for Older Adults: Participatory Action Research with Older Adults

Point-by-point responses to Reviewer’s comments and suggestions

For the abstract, is it possible to provide a more user-friendly definition of Frailty? The journal is “IJERPH” so the readers may be clinicians such as nurses or therapists. The current definition in the abstract is very bench research-sounding and not translational.

Response: Thank you for helpful advice. We have addressed a more user-friendly definition of Frailty as follows:

Abstract: Line 1

Frailty is an age-related condition characterized by a decline in physical capacity with an increased vulnerability to stressors.

 The introduction: There is a lack of why this device was created and what issues/limiting factors this device is intending to address. Please add to, and rephrase, frailty to describe a more clinician-friendly definition and take out the reference from 2001. The only limiting factor of frailty that I’m understanding is “physiological capacity due to organ systems.”   The literature review is currently not sufficient for a manuscript to be published, it is lacking critical support to design this device based on a review of technology that is already on the market to address the issue. I’m not understanding what is current, or lacking on the market, to address physiological capacity.  What exactly is this: balance, heart rate, blood pressure, loss of muscle mass, falls?  What is the support in the literature of the development of devices that are technologically sound, feasible, and have preliminary efficacy to address this (outside of Japan), and other issues, of the older adult?  There should also be in the intro any background literature on older adult reports of their acceptance of telehealth/online health sessions, and provide the background work of others to support this device. This should guide then your research question, which the article is also lacking: Please provide a clear research question.

Response: Thank you for the insightful comment. We have addressed the definition of frailty and its characteristics with several recent references (ref.3~8) to be a more clinician-friendly (line 42-50).

“Frailty exposes the individual to a greater risk of multiple adverse health outcomes such as loss of mobility, falls/fractures, hospitalization, and early mortality [3-7]. Three important factors have been suggested in the concept of frailty. First, frailty is multidimensional with physical, cognitive, and social factors. Second, although its prevalence does increase with age, frailty is an extreme consequence of the normal ageing process. Third, frailty is reversable, which means that an individual can prevent or slow its progression [8]. Therefore, effective strategies that target the prevention and management of frailty in an ageing population will probably reduce the condition’s burden at l both the individual and the health system level.”

  1. Clegg, A.; Young, J.; Iliffe, S.; Rikkert, M.O.; Rockwood, K. Frailty in elderly people. Lancet 2013, 381, 752–62. DOI:10.1016/S0140-6736(12)62167-9
  2. Hoogendijk, E.O.; Muntinga, M.E.; van Leeuwen, K.M.; van der Horst, H.E.; Deeg, D.J.H.; Frijters, D.H.M.; Hermsen, L.A.H.; Jansen, A.P.D.; Nijpels, G.; van Hout, H.P.J. Self-perceived met and unmet care needs of frail older adults in primary care. Arch Gerontol Geriatr 2014, 58, 37–42. DOI:10.1016/j.archger.2013.09.001.
  3. Vermeiren, S.; Vella-Azzopardi, R.; Beckwée, D.; Habbig, A.-K.; Scafoglieri, A.; Jansen, B.; Bautmans, I.; Bautmans, I.; Verté, D.; Beyer, I.; et al. Frailty and the prediction of negative health outcomes: a meta-analysis. J Am Med Dir Assoc 201617, 1163.e1–1163.e17. DOI:10.1016/j.jamda.2016.09.010.
  4. Junius-Walker, U.; Onder, G.; Soleymani, D.; Wiese, B.; Albaina, O.; Bernabei, R.; Marzetti, E. The essence of frailty: a systematic review and qualitative synthesis on frailty concepts and definitions. Eur J Intern Med 201856, 3–10. DOI:10.1016/j.ejim.2018.04.023.
  5. Yang, X.; Lupón, J.; Vidán, M.T.; Ferguson, C.; Gastelurrutia, P.; Newton, P.J.; Macdonald, P.S.; Bueno, H.; Bayés‐Genís, A.; Woo, J.; et al. Impact of frailty on mortality and hospitalization in chronic heart failure: a systematic review and meta-analysis. J Am Heart Assoc 20187, e008251. DOI:10.1161/JAHA.117.008251.
  6. Markle-Reid, M.; Browne, G. Conceptualizations of frailty in relation to older adults. J Adv Nurs 200344, 58–68. doi:10.1046/j.1365-2648.2003.02767.x.

We also performed a literature review of technology, especially for online frailty assessments and found three available references (ref. 22-24). With study description and issues to solve in these recent studies, we clarified the aim of our study at line 76-87. Please see the revised text below. 

“Recently, studies have reported online frailty assessments for older adults [22, 23] or health professionals [24]. For example, in community settings, using online version of the FRAIL scale (a simple questionnaire including five items: fatigue, resistance, ambulation, illness, and loss of weight), participants aged 60 years or older were able to understand the questions and answer using tablets [22]. Significant correlation with SARC-F was also observed. However, another study showed that only 35% older adults accepted a digital approach for conducting health assessment or accessing assessment results [22].

To be acceptable for older adults based on the established onsite FC program, we co-designed/co-developed an online FC application with FC supporters through participatory action research (PAR). In this study, we aimed to examine the usability of online FC focusing on reliability defined as consistency of pre-existing onsite FC, and online communication. We also validated the online FC scores with those of onsite FC.”  

  1. Yu, R.; Tong, C.; Leung, G.; Woo, J. Assessment of the validity and acceptability of the online FRAIL scale in identifying frailty among older people in community settings. 2021, 145, 18–23. DOI: 10.1016/j.maturitas.2020.12.003.
  2. Lee, J.S.; Chew, J.; Lim, Y.R.; Ng, W.K.G.; Yeo, A.J.P.; Ong, L.T.J.; Chan, M.P.C.; Lim, W.S.; Beauchet, O. Validating the Centre of Excellence on Longevity Self-AdMinistered (CESAM) Questionnaire: An Online Self-Reported Tool for Frailty Assessment of Older Adults. J Am Med Dir Assoc. 2022, 23, 1984.e1–1984.e8. DOI: 10.1016/j.jamda.2022.06.031.
  3. Haddad, T.; Mulpuru, S.; Salter, I.; Hladkowicz, E.; Des Autels, K.; Gagne, S.; Bryson, G.L.; McCartney, C.J.L.; Huang, A.; Huang, S.; et al. Development and evaluation of an evidence-based, theory-grounded online Clinical Frailty Scale tutorial. Age Ageing. 2022, 51, afab258. DOI: 10.1093/ageing/afab258.

Please also add if this was approved by an ethics board and when subjects consented to their participation.

Response: Thank you for your valuable comment. As per the reviewer’s suggestion, we have included the approval by an ethics board of the University of Tokyo with approval number in the materials and methods section (line 94-97). Please see the revised text below.

“This study protocol was reviewed and approved by the University of Tokyo institutional review committee (approval number: 21-190).Written informed consent was obtained form all FC supporters and participants by researchers beforey interview or recruitment.”

Line 109: What were the methods that guided your qualitative portion of this? i.e. content analysis, thematic analysis, phenomenology, grounded theory?

Response: As the reviewer’s suggestion, we have addressed for qualitative analysis (grounded theory) in the materials and methods section at line 164-166 with relevant references (ref.31-33). Please see the revised text below.

“We explored the phenomenon of challenges that participants gained from their experience using the application. Therefore, we followed the research of application development and action research based on a grounded theory study [31-33].”

  1. Glaser, B.G.; Strauss, A.L. The discovery of grounded theory: strategies for qualitative research, Aldine Pub. Co.: Chicago, 1967.
  2. Gerlach, J. P.; Cenfetelli, R. T. Constant Checking Is Not Addiction: A Grounded Theory of IT-Mediated State-Tracking. MIS Quarterly, 2020, 44(4), 1705-1731.
  3. Hand, C.; Rudman, D.L.; McGrath, C.; Donnelly, C.; Sands, M. Initiating participatory action research with older adults: lessons learned through reflexivity. Can J Aging. 2019, 38, 512–520. DOI:1017/S0714980819000072.

There is no description of the General Self-Efficacy Scale questionnaire. How are the scores compiled, what are the ranges of scores, and what are the cut-off scores for interpretation? Is this published?  What are the psychometric properties? Reliability, validity, MCID? If not and it was created, did it go through a pilot phase? 

Response: Thank you for your valuable comment. As per your suggestion, we have included a sufficient description of the General Self-Efficacy Scale questionnaire at line 172-178. Please find the revised text below.

“To examine whether FC supporters were empowered by PAR, we assessed self-efficacy using a validated 16-item questionnaire on the general self-efficacy scale (GSES) consisting of three categories: positive behavior, non-anxiety about failure, and social positioning of ability [34]. The total possible score on the GSES is 16 points. Based on results of a previous study in the Japanese population (n=278), it was suggested that 10-16 points indicate higher self-efficacy, 4-9 points indicate standard self-efficacy, and 0-3 points indicate lower self-efficacy [34].”

In addition, since we sought to examine the empowerment of FC supporters through PAR using this questionnaire, we have addressed this purpose in lines 245-247. Please find the revised text below.

“In the co-design and co-development of the online FC application, we investigated whether the engagement of FC supporters in PAR was empowered, using a validated self-efficacy assessment.”

The same comments above apply to the SUS which requires sufficient descriptions. It appears in your graph that 70 is a cut-off of SUS acceptability, but this needs to be stated in the text if there is literature to support this. This usually is listed as the “Measurement” section of a Methods section.

Response: With respect to the SUS, we also have addressed detail description in the materials and methods section and a cut-off 68 points is mentioned at line 182-189 with relevant references (ref. 36-38). Please find the revised text below.

“For quantitative data, we scored the system usability scale (SUS). The SUS has been extensively used in previous user research studies and demonstrated good psychometric properties [36]. Better usability was indicated by higher SUS scores, ranging from 0 to 100 [37]. SUS scores of 68 is center of the Sauro-Lewis curved grading scale, which is one of analyzing point of usability [37, 38]. SUS consists 10 items, each with five steps anchored with "Strongly Disagree" and "Strongly Agree." It is a mixed-tone questionnaire in which the odd-numbered items have a positive tone and the even-numbered items have a negative tone.”

  1. Brooke, J. SUS: a ‘quick and dirty’ usability scale. Usability Eval Ind. 1996, 189, 4–7.
  2. Lewis, J.R. The system usability scale: past, present, and future. In J Hum Comput Interact. 2018; 34:577-590. Doi: 1080/10447318.2018.1455307.
  3. Lew, J.R., Sauro, J. Item benchmarks for the system usability scale. Journal of Usability Studies. 2018;13:158-167.

Discussion: There is also not a sufficient discussion of your interpretations of the GSES and the SUS scores if the cut-offs were met for an older adult.

Response: Thank you for your valuable comment. As per your suggestion, we have addressed the interpretation of the GSES and SUS at line 325-341 in the discussion section adding new references (ref.44, 45). Please find the revised text below.

    “Intriguingly, regarding self-efficacy of FC supporters, we found that an item for the social positioning of ability, significantly changed after mock test and a trend of improvement in the non-anxiety of failure was seen. These results suggested that the FC supporters were able to strengthen their own social positioning of ability and relieve anxiety about failure through PAR in the design/development of the application. This highlights the promotive effects on self-efficacy for FC supporters, although the average total score did not significantly change because total score at before time point was high enough. Furthermore, consistent with a recent study on PAR [44], the empowerment of the FC supporters equips them with the confidence to operate the online FC, which subsequently could enhance the usability for all participants.

In this study, we found that the average score of SUS for online FC application are 70.2±10.3 which ranged in marginally high of acceptability. Consistently, it is demonstrated that SUS scores of community-dwelling older adults (70.9 ±5.6 years) are 68 in the recent pilot study for the system for assessment and intervention of frailty [45]. Although, SUS is a standardized questionnaire designed to assess perceive usability [36], there is a lack of studies exploring the usability of technology especially for older adults. Further investigations should be needed by large-scale randomized controlled trial. “

  1. Ros-Sanchez T, Lidon-Cerezuela MB, Lopez-Benavente Y, Abad-Corpa E. Promoting empowerment and self-care in older women through participatory action research: Analysis of the process of change. J Adv Nurs. 2023, 79, 2224–2235. DOI: 10.1111/jan.15573.
  2. Tan, R.S., Goh, E.F., Wang, D., Chan, RCL., Zeng, Z., Yeo, A., Pek, K., Kua, J., Wong, W.C., Shen, Z., Lim, W.S. Effectiveness and usability of the system for assessment and intervention of frailty for community-dwelling pre-frail older adults: A pilot study. Front Med (Lausanne). 2022;9:955785. doi: 10.3389/fmed.2022.955785. 

In the discussion, this is the first mention that this is a feasibility study, which cannot be introduced in a discussion.  If this is an aim, add it to the research question and how feasibility was planned to be measured/assessed. If not, please take out the phrasings of feasibility.

Response: Thank you for your valuable comment. As you have pointed out, since we aimed to examine usability and validate online FC, we have removed the phrasings of feasibility and have modified as follow (line 289-294): 

“In this study, we developed an online FC application for community-dwelling older adults delivered by video conferencing, during the COVID-19 pandemic. To enhance its acceptability in older adults through PAR, we co-designed/co-developed it with stakeholders named FC supporters who are older community-resident volunteers facilitating the pre-existing onsite FC. This study aimed to examine the usability and validity of the online FC application.”

Comments on the Quality of English Language

Some language is too technical for the clinician readership and lacks translational application.  I asked the authors to address this.

Response: Thank you for your valuable comment. As per your suggestion, we have used more reader-friendly language and added detailed explanations throughout the text.

Reviewer 2 Report

Dear authors,

thank you for submitting this manuscript and the interesting study. Your article is good, but you need to provide the reader with more detail. In your introduction you will need to set the scene better. For example I would like to understand better what type of participatory design philosophy you subscribe to. As you have currently reported it - I consider this as co-design under the human-centred design paradigm and you applied 'traditional' approaches to capturing user opinions (questionnaires, scales, focus group) - pure participatory research shifts power relations and I did not get from this report of your research that you have empowered your co-designers. (you mention increased self-efficacy for the FC supporters, but this is not the same as having a say in the outset of the project i.e. what type of technology - how should it be delivered). 

I further lacked an understanding of how the onsite appointments work - do they work in groups, so I'm a bit confused why you would have 6 people online tool. Introducing a digital version of a existing system offers opportunities to improve what exists... in this study i missed an indication (of course more research is needed) for the bespoke benefits of carrying the assessment out online (e.g. no risks of falling going to the appointment) or to list the disadvantages (e.g. not going to meet people in person, appointments give some people a reason to leave the house)

I further would have liked to see this project situated in a range of other research / project. Codesign has been successfully used in design for health applications - there is plenty of literature.  your article lacks references in the beginning of the document. for example  there is no reference to what type of participatory action research in the method section.

Please add how you have retrieved ethical consent from the participants.

the use of diagrams is great. your limitations should include a consideration for your cultural context, and how generalisable it is.   

here are some further specific comments:

At the moment I’m not quite clear how this assessment is carried out.  Even though you describe this in 85ff I expected multiple choice and tick boxes rather than writing. I also wonder why you would do this FC application in a group setting rather than 1-to-1 or does the group up to 6 people consist of key person, family members and carers?

Finding out that they should use a stylus, enlarge the font and induce a button pressing simulator is not something that would need 52 people to experience it before figuring this out from a design perspective

Table 2: code “reliability” seem to include a wide array of functionality – please describe before (in line 109ff) what you consider as ,reliability, … I understood it as a system not crashing

figure 1 - I'm confused about the recruitment box without extra words, please add "community dwelling participants" to it

figure 1 b) - am also confused about the amount of participants taking part in the online check - I expected it to be one person at the time maybe together with a supporter, but why are there 6 people? who is the facilitator and what exactly does this person do? 

line 100 "participatory survey method to conduct the research" - can you give us a reference for this method? 

lines 109-112 - what approach is this to analysis - please spell out and provide a reference 

line 114 why is the self-efficacy of the FC supporters assessed? why do the supporters need self-efficacy, i thought it would need to be the older person themselves

lines 205ff - so what is the significant association between onsite and online results? do you mean that results were the same for participants taking the test onsite or online? (this should be the case anyway.... or were you doubting this)

220 you need to define 'reliability'

252ff- what do you mean with "audio-video quality" should be improved to help prevent frailty, in terms of social isolation and loneliness? How can better audio / video quality improve the prevention of frailty and only when people connect with video & audio social isolation and loneliness maybe addressed, but otherwise you will find that there are huge difference between the two and just by increase the number of contact points / social interaction does not mean one would feel less lonely

227 - there are plenty of examples of co-design and health care interventions (see suggested references below)

260 - i thought you wanted to design this as an alternative to onsite... (due to the lock down circumstances)

line 269 - feasibility and acceptability - please add usability 

Useful references for more co-design for health care applications: Ting, K. L. H., Dessinger, G., & Voilmy, D. (2020). Examining usage to ensure utility: Co-design of a tool for fall prevention. IRBM41(5), 286-293.c

Noorbergen, T. J., Adam, M. T., Roxburgh, M., & Teubner, T. (2021). Co-design in mHealth systems development: insights from a systematic literature review. AIS Transactions on Human-Computer Interaction13(2), 175-205.

Sanz, M. F., Acha, B. V., & García, M. F. (2021). Co-design for people-centred care digital solutions: a literature review. International Journal of Integrated Care21(2).

Author Response

Open Review

Point-by-point responses to Reviewer’s comments and suggestions

Dear authors,

thank you for submitting this manuscript and the interesting study. Your article is good, but you need to provide the reader with more detail. In your introduction you will need to set the scene better. For example I would like to understand better what type of participatory design philosophy you subscribe to. As you have currently reported it - I consider this as co-design under the human-centred design paradigm and you applied 'traditional' approaches to capturing user opinions (questionnaires, scales, focus group) - pure participatory research shifts power relations and I did not get from this report of your research that you have empowered your co-designers. (you mention increased self-efficacy for the FC supporters, but this is not the same as having a say in the outset of the project i.e. what type of technology - how should it be delivered). 

Response: Thank you for the insightful comments. As per the reviewer’s suggestions, to better understand the scene of FC and the role of FC supporter as a stakeholder, we have added detail descriptions in the introduction section (lines 51-65). Please find the revised text below.

“In Japan, to screen frailty in community-dwelling older adults, we developed a frailty check (FC) program. This program is based on evidence from a large-scale longitudinal cohort study in older adults in Kashiwa City who did not require long-term care needs [9]. The FC program consists of a self-assessment for sarcopenia (Yubi-wakka test) [10] and an 11-item self-reporting questionnaire. Particularly, this simple check version is designed for self-awareness to motivate people to change their lifestyle to incorporate good nutrition [11], physical activity [12, 13], and social participation [14]; three pillars from the perspective of preventing frailty. Notably, the FC program is characterized by a citizen-centered action handled by FC supporters who are community-dwelling older volunteers. The FC program was conducted in groups at public halls and community centers easily accessed by older adults. FC supporters encourage participants to improve their lifestyle and practice for frailty prevention [15]. From April 2015 to February 2020, a total of 8,855 community-dwelling older adults participated in the FC program in 47 local government prefectures. Thus, FC supporters are important stakeholders of frailty prevention in the community.”

We have also addressed the participatory design in the study design of method section (line 101-106). Please find the revised text below.

To develop and implement the online FC application, we considered PAR to be the most appropriate method for several reasons: (1) frailty screening and prevention are critical issues in the community, (2) its participatory nature involves FC supporters in the research from the beginning to the end, and (3) it involves research in action. However, there is limited research using PAR as a technological tool for frailty screening, especially in older adults [25, 26].

Furthermore, we have included insights of our approach by PAR in the discussion (line 296-304). Please find the revised text below.

“To achieve our purpose, we utilized a PAR approach [33, 40, 41], consisting of focus group interviews and a mock test. Given that the FC program is an independent civic activity and FC supporters were not simply users but also operators who should facilitate both onsite and online FC, they were involved in all steps of this study from the application design, e.g., what type of technology is acceptable for older adults and how should it be delivered to participants to the step of implementation of online FC with participants, which are finally attributed to their empowerments. Based on this observation, it is conceivable that a participatory design might be a useful methodology when developing digital healthcare interventions for older adults.”

I further lacked an understanding of how the onsite appointments work - do they work in groups, so I'm a bit confused why you would have 6 people online tool. Introducing a digital version of a existing system offers opportunities to improve what exists... in this study i missed an indication (of course more research is needed) for the bespoke benefits of carrying the assessment out online (e.g. no risks of falling going to the appointment) or to list the disadvantages (e.g. not going to meet people in person, appointments give some people a reason to leave the house)

Response: Thank you for your valuable comment. As per your suggestion, we have included a detailed description for onsite FC in the introduction (line 51-65) and further, included the reason for a 6 person layout and how to perform the online FC in the materials and methods (line 140-153). Please find the revised text below.

<In the introduction>

“In Japan, to screen frailty in community-dwelling older adults, we developed a frailty check (FC) program. This program is based on evidence from a large-scale longitudinal cohort study in older adults in Kashiwa City who did not require long-term care needs [9]. The FC program consists of a self-assessment for sarcopenia (Yubi-wakka test) [10] and an 11-item self-reporting questionnaire. Particularly, this simple check version is designed for self-awareness to motivate people to change their lifestyle to incorporate good nutrition [11], physical activity [12, 13], and social participation [14]; three pillars from the perspective of preventing frailty. Notably, the FC program is characterized by a citizen-centered action handled by FC supporters who are community-dwelling older volunteers. The FC program was conducted in groups at public halls and community centers easily accessed by older adults. FC supporters encourage participants to improve their lifestyle and practice for frailty prevention [15]. From April 2015 to February 2020, a total of 8,855 community-dwelling older adults participated in the FC program in 47 local government prefectures. Thus, FC supporters are important stakeholders of frailty prevention in the community.”

<In the materials and methods>

   “The online FC was conducted via the FC application using a tablet device by joining a video conference room with six seats (FC supporter: participants= 1~2 : 1~5). This ratio was intended to enable FC supporters to adequately attend to the participants (Fig. 1b). In this video conference room, an FC supporter could share the questionnaire with the participants on screen and grasp the state of participants responses. Following the FC supporters’ instructions, participants would input data or answer questions using an electronic pencil.

The online FC program consisted of three sections: basic information (age, sex, body weight, height), Yubi-wakka test (sarcopenia test with video instruction [10]) and an 11-item FC questionnaire (yes/no, validated questionnaires including questions on nutrition, oral and physical function, and social activity). Inputted data was summarized at the end which the participants could confirm prior to submission. This design was also used in the onsite FC program. Each section of the 11-item questionnaire is addressed in Table S1 and a movie of the online FC implementation is also supplied as Video S1 (in Japanese).”

Furthermore, as the reviewer’s suggestion, we have addressed benefits and disadvantage of online assessment with our goal, a continuous and effective FC by hybrid system in community in the discussion (line 316-320). Please find the revised text below.

“In this study, we suggested the use of continuous and effective FC in the community by a hybrid system linking onsite (every 6 month) and online FC (e.g., once or twice between 6 months) by understanding benefits (e.g., no risks of falling going to the appointment) and disadvantages (e.g., not going to meet people in person, appointments give some people a reason to leave the house) of online FC.”

I further would have liked to see this project situated in a range of other research / project. Codesign has been successfully used in design for health applications - there is plenty of literature.  your article lacks references in the beginning of the document. for example  there is no reference to what type of participatory action research in the method section.

Response: Thank you for your valuable comment. As you have pointed out, we have included our approach of PAR in the Methods section (line 101-106) with two new references (ref. 25, 26). Please find the revised text below.

“To develop and implement the online FC application, we considered PAR to be the most appropriate method for several reasons: (1) frailty screening and prevention are critical issues in the community, (2) its participatory nature involves FC supporters in the research from the beginning to the end, and (3) it involves research in action. However, there is limited research using PAR as a technological tool for frailty screening, especially in older adults [25, 26].”

  1. Dickson, G. Aboriginal Grandmothers' experience with health promotion and participatory action research. Qual Health Res, 2000, 10, 188–213. DOI:10.1177/104973200129118363.
  2. Kingery, F.P.; Naanyu, V.; Allen, W.; Patel, P. Photovoice in Kenya: Using a community-based participatory research method to identify health needs. Qual Health Res, 2016, 26, 92–104. DOI:10.1177/1049732315617738

Please add how you have retrieved ethical consent from the participants.

Response: Thank you for your valuable comment. As your suggestion, we have included the method of ethical consent approval in the materials and methods section (line 94-97). Please find the revised text below.

“This study protocol was reviewed and approved by the University of Tokyo institutional review committee (approval number: 21-190).Written informed consent was obtained form all FC supporters and participants by researchers beforey interview or recruitment.”

the use of diagrams is great. your limitations should include a consideration for your cultural context, and how generalisable it is.   

 Response: Thank you for your valuable comment. As per your suggestion, we have addressed the cultural or generalizable limitation in the Limitations section (line 364-366). Please find the revised text below.

“Our design/development was limited to Japanese older adults. To build a generalizable system, an established protocol of PAR involving community stakeholders is needed.”

here are some further specific comments:

 At the moment I’m not quite clear how this assessment is carried out.  Even though you describe this in 85ff I expected multiple choice and tick boxes rather than writing. I also wonder why you would do this FC application in a group setting rather than 1-to-1 or does the group up to 6 people consist of key person, family members and carers?

Response: Thank you for your valuable comment. To better explain our FC system, first, we have included a detailed description of both onsite (line 51-65) and online FC (line 140-153) assessment.  Our FC system is designed to place importance to communication among old adults in community, which is attributed to building a sustainable citizen-centered action. Thus, we settled for an online FC assessment in group, like the onsite FC. We have addressed the meaning of a group setting and composition of member in the detailed description.

 Finding out that they should use a stylus, enlarge the font and induce a button pressing simulator is not something that would need 52 people to experience it before figuring this out from a design perspective

Response: As the reviewer’s suggestion, we have addressed our perspective in detail (line 233-241). Please find the revised text below.

“Furthermore, several opinions on user-friendliness were collected and improvements based on this feedback were implemented: 1) using a stylus, 2) enlargement of font size, and 3) introducing a button pressing simulator. In particular, we found that including a stylus significantly improved usability by solving dry fingers and touch sensitivity issues, especially for older adults. We also received suggestions on font size enlargement, improving readability. A button press simulator was also included for participants new to tablets, bridging the gap between physical interfaces and touch-based interaction for a smoother transition. Our study involved 52 participants, ensuring a diverse sample size to capture various perspectives and usability issues.”

 Table 2: code “reliability” seem to include a wide array of functionality – please describe before (in line 109ff) what you consider as ,reliability, … I understood it as a system not crashing

Response: Thank you for your valuable comment. As per the reviewer’s suggestion, we have addressed the definition of reliability in this study at the introduction section (line 86). Please find the revised text below.

“In this study, we aimed to examine the usability of online FC focusing on reliability defined as consistency of pre-existing onsite FC, and online communication.”

 figure 1 - I'm confused about the recruitment box without extra words, please add "community dwelling participants" to it

Figure 1 b) – am also confused about the amount of participants taking part in the online check – I expected it to be one person at the time maybe together with a supporter, but why are there 6 people? Who is the facilitator and what exactly does this person do? 

Response: Thank you for your valuable comment. As you have pointed out, to better understand, we have added community-dwelling participants (a), inserted a picture of a FC supporter who played a role of facilitator and his role (b).

lines 109-112 - what approach is this to analysis - please spell out and provide a reference 

Response: As the reviewer’s suggestion, we have addressed for qualitative analysis (grounded theory) in the materials and methods section at line 164-166 with relevant references (ref.31-33). Please see the revised text below.

“We explored the phenomenon of challenges that participants gained from their experience using the application. Therefore, we followed the research of application development and action research based on a grounded theory study [31-33].”

  1. Glaser, B.G.; Strauss, A.L. The discovery of grounded theory: strategies for qualitative research, Aldine Pub. Co.: Chicago, 1967.
  2. Gerlach, J. P.; Cenfetelli, R. T. Constant Checking Is Not Addiction: A Grounded Theory of IT-Mediated State-Tracking. MIS Quarterly, 2020, 44(4), 1705-1731.
  3. Hand, C.; Rudman, D.L.; McGrath, C.; Donnelly, C.; Sands, M. Initiating participatory action research with older adults: lessons learned through reflexivity. Can J Aging. 2019, 38, 512–520. DOI:1017/S0714980819000072.

 line 114 why is the self-efficacy of the FC supporters assessed? why do the supporters need self-efficacy, i thought it would need to be the older person themselves

Response: Thank you for your valuable comment. As per your suggestion, we have addressed how the self-efficacy of the FC supporters is assessed in the Materials and Methods section (lines 172-178).

“To examine whether FC supporters were empowered by PAR, we assessed self-efficacy using a validated 16-item questionnaire on the general self-efficacy scale (GSES) consisting of three categories: positive behavior, non-anxiety about failure, and social positioning of ability [34]. The total possible score on the GSES is 16 points. Based on results of a previous study in the Japanese population (n=278), it was suggested that 10-16 points indicate higher self-efficacy, 4-9 points indicate standard self-efficacy, and 0-3 points indicate lower self-efficacy [34].”

lines 205ff - so what is the significant association between onsite and online results? do you mean that results were the same for participants taking the test onsite or online? (this should be the case anyway.... or were you doubting this)

Response: Thank you for the insightful comment. To build a hybrid system linking onsite and online FC as addressed in the discussion section (lines 316-320), we expected similar results in both FCs. With this comparison, we found the significant association between onsite and online results.

“In this study, we suggested the use of continuous and effective FC in the community by a hybrid system linking onsite (every 6 month) and online FC (e.g., once or twice between 6 months) by understanding benefits (e.g., no risks of falling going to the appointment) and disadvantages (e.g., not going to meet people in person, appointments give some people a reason to leave the house) of online FC.”

220 you need to define 'reliability'

Response: Thank you for your valuable comment. As per your suggestion, we have addressed the definition of reliability in this study in the introduction section (line 86).

“In this study, we aimed to examine the usability of online FC focusing on reliability defined as consistency of pre-existing onsite FC, and online communication.”

252ff- what do you mean with "audio-video quality" should be improved to help prevent frailty, in terms of social isolation and loneliness? How can better audio / video quality improve the prevention of frailty and only when people connect with video & audio social isolation and loneliness maybe addressed, but otherwise you will find that there are huge difference between the two and just by increase the number of contact points / social interaction does not mean one would feel less lonely

Response: Thank you for your valuable comment. We apologize for the confusion. We have provided examples for audio-video quality, to enhance communication (lines 348-351). In addition, we have addressed the importance of social network as a first step of frailty progression with reference to the structural equation modeling analysis (ref. 46). Please find the revised text below.

“For example, audio-video quality (e.g., dim lighting, small voice) should be improved to enhance communication, particularly in terms of social isolation and loneliness, considered as the first step of frailty [46].”

  1. Tanaka, T.; Son, B.; Lyu, W.; Iijima, K. The impact of social engagement on the development of sarcopenia among community-dwelling older adults: A Kashiwa cohort study. Geriatr Gerontol Int 2022, 22, 384–391.DOI: 10.1111/ggi.14372. 

227 - there are plenty of examples of co-design and health care interventions (see suggested references below)

Response: Thank you for your valuable comment and the useful references. We have included these references in the Materials and Methods section (line 107-108, ref. 27-29). Please find the revised text below.

“To co-design the application, referring to recent studies [27-29], focus group interviews and mock tests were conducted.”

  1. Lan Hing Ting, K.; Dessinger, G.; Voilmy, D. Examining Usage to Ensure Utility: Co-Design of a Tool for Fall Prevention. IRBM 2020, 14, 286–293. DOI:10.1016/j.irbm.2020.03.001
  2. Sanz, M.F.; Acha, B.V.; García, M.F. Co-Design for People- Centred Care Digital Solutions: A Literature Review. Int J Integr Care. 2021, 21, 16, 1–17. DOI: 10.5334/ijic.5573
  3. Noorbergen, T.; Adam, M.; Roxburgh, M.; Teubner, T. Co-Design in mHealth Systems Development: Insights From a Systematic Literature Review. ACM Trans Comput Hum Interact 202113, 175–205. DOI:10.17705/1thci.00147.

260 - i thought you wanted to design this as an alternative to onsite... (due to the lock down circumstances)

Response: Thank you for your valuable comment. We apologize for the confusion. We have modified the description of aim of online FC development (for hybrid system) to better understand (lines 316-320, 354-357).  Please find the revised text below.

“In this study, we suggested the use of continuous and effective FC in the community by a hybrid system linking onsite (every 6 month) and online FC (e.g., once or twice between 6 months) by understanding benefits (e.g., no risks of falling going to the appointment) and disadvantages (e.g., not going to meet people in person, appointments give some people a reason to leave the house) of online FC.”

“With this observation, it is suggested that the online FC application could be a reliable tool of onsite FC among older participants, for sustained monitoring of frailty status with hybrid system and further use for emergency situations like the COVID-19 pandemic.”

line 269 - feasibility and acceptability - please add usability 

Response: Thank you for your valuable comment. As per your suggestion, we have added usability (line 360-361). Please find the revised text below.

“The primary focus was on developing the application and conducting a preliminary evaluation of its acceptability, reliability, and usability.”

 Useful references for more co-design for health care applications: 

Ting, K. L. H., Dessinger, G., & Voilmy, D. (2020). Examining usage to ensure utility: Co-design of a tool for fall prevention. IRBM41(5), 286-293.c

Noorbergen, T. J., Adam, M. T., Roxburgh, M., & Teubner, T. (2021). Co-design in mHealth systems development: insights from a systematic literature review. AIS Transactions on Human-Computer Interaction13(2), 175-205.

Sanz, M. F., Acha, B. V., & García, M. F. (2021). Co-design for people-centred care digital solutions: a literature review. International Journal of Integrated Care21(2).

Round 2

Reviewer 1 Report

Thank you for addressing my concerns about adding more background evidence and details in the methods as to what gaps in the literature this work is fulfilling.